# Disparity in Spatial Access to Public Daycare and Kindergarten across GIS-Constructed Regions in Seoul, South Korea

**Hyunjoong Kim and Fahui Wang \***

Department of Geography & Anthropology, Louisiana State University, Baton Rouge, LA 70803, USA;
hkim57@lsu.edu

\* Correspondence: fwang@lsu.edu

**Abstract:** In developed countries with decreasing fertility rates, the provision of public daycare and kindergarten (PDK) is considered to be an important policy for boosting national birth rates. Since PDK is free, its spatial accessibility becomes the most critical factor for parents in choosing the service. The study uses the popular two-step floating catchment area model (2SFCA) to analyze the spatial accessibility of PDKs at a 100 m × 100 m cell level in Seoul, South Korea. A GIS-automated regionalization method, Mixed-Level Regionalization (MLR), is employed to divide the study area into homogenous regions based on a concentrated disadvantage index (CDI). The analysis then proceeds to examine the disparity of PDK accessibility across these constructed regions. The result empowers parents to be informed of the access of PDKs in their current neighborhoods or to look for neighborhoods with adequate access. Several policy measures are proposed for improving overall accessibility of PDKs and more so for underserved populations.

**Keywords:** spatial accessibility; public daycare and kindergarten (PDK); two-step floating catchment area (2SFCA) method; mixed-level regionalization (MLR); concentrated disadvantage index (CDI); Seoul

## 1. Introduction

More economically developed countries have low fertility rates, which, along with the aging population, may adversely affect public finances and standards of living [1]. In 2018, South Korea's total fertility rate dropped to 0.96 [2], the lowest among the Organization for Economic Cooperation and Development (OECD) countries. Among various policy alternatives to cope with the challenge, women's labor participation has attracted much attention, since it has been shown to lower birth rates [3,4]. Therefore, a wide range of family-related policies has been proposed to encourage women to stay in the workforce while caring for their children. One of these policies focuses on improving accessibility of childcare [5–7]. One recent effort in South Korea is the expansion of public daycares and kindergartens (hereafter referred to as "PDKs") in lieu of subsidizing private childcare services [8]. In light of this ongoing campaign, it is important to understand which parents and locations receive more benefits, and which parents and locations are left behind.

Many positive effects of daycares and kindergartens have long been noted in the literature. Preschools are recognized for providing developmental programs that enhance physical activity [9]. Therefore, spending greater amounts of time in early childhood centers helps to prevent childhood obesity and other chronic diseases [10]. They also play an important role in fostering the sociality of children. Those with fewer social skills acquired in their childhoods are more likely to quit school, engage in criminal activities, or have difficulties finding a job [11,12]. Preschools also contribute

to developing critical skills, including language [13]. Children who have sufficient language skills in preschool are likely to have good overall learning skills [14]. Preschools play a very positive role in children's growth and should be valued from a comprehensive perspective beyond childcare. Accessibility for childcare is essential for children's convenience, health, and safety. Since the 2000s, there has been an increasing interest in studying the accessibility of childcare facilities [15–20]. Eliminating various barriers for childcare access is part of the public's social responsibility.

In South Korea, PDK accessibility is a key decision factor for parents on whether and where to enroll their children [21]. It matters even more for socially disadvantaged groups [16,22,23]. A longer distance incurs more financial cost and requires more time commitment, which can affect a mother's job selection [20]. The shortage of PDKs may force people to rely on private daycare that may not be feasible for the socially vulnerable [24]. In essence, improved accessibility of PDKs helps promote women's labor participation [5,20,24–26].

Spatial accessibility is defined as the convenience of reaching a particular service by overcoming spatial barriers (e.g., distance and transportation means) between the service providers and residents [27]. An important endeavor in the analysis of childcare accessibility is the pursuit of accurate measurement. Often, due to the lack of quality data, previous studies employed aggregate data to measure the demand side, such as numbers of children or women at child-bearing ages in large geographic units. Such studies did not reveal the full challenges faced by parents in seeking daycare [25]. Researchers should employ micro-data for securing more accurate measures of accessibility in neighborhood facilities and reducing aggregation errors [28]. A fine spatial resolution for the demand data is especially critical as parents are very sensitive to travel distance for childcare and usually have a small search radius for such facilities. This research uses a fine-grained population data set recently constructed by the Korean government. It covers a population of all age groups in 100 m × 100 m cell sizes, derived from the nation's census data in 2015. This study measures the demand as children of ages 0–6 years old.

The disparity in access to resources or public services between the haves and have nots is a constant theme of research inquiry inspired by the social justice theory [29] and others e.g., [30–32]. Since spatial accessibility is a location-based measure, various demographic groups in one neighborhood share the same accessibility value, but the levels of concentration by certain groups vary across neighborhoods. Therefore, some studies assess disparity across subpopulations by testing whether one demographic group is disproportionally represented in areas of below-average accessibility scores [33,34]. This study proposes a new framework for assessing the issue. We employ the state-of-art GIS-automated regionalization method (see Section 3.2) to divide the study area into regions that are spatially contiguous and homogenous in attributes, and then examine the disparity of accessibility across these constructed regions. Since these regions are derived in a way that maximizes homogeneity within each area, they are distinctive in socio-economic structure, similar to clusters used in urban social area analysis [35]. However, social area analysis is based on cluster analysis of attributes only and yields areas that are not necessarily spatially contiguous.

Major contributions of this study are three-fold. First, it analyzes the accessibility of PDKs more accurately by adopting a fine spatial resolution data. Secondly, by constructing comparative regions that are more coherent in socio-economic structure than administrative units, the subsequent analysis of geographic disparity of accessibility is better equipped to identify where and who experience poorer access. In essence, the GIS-automated regionalization method helps pinpoint the concentration areas of population subgroups. Lastly, future policies on the spatial planning of PDKs can leverage the approach and the results of this study to increase effectiveness. When a proposed PDK location-allocation scheme is contemplated, one can follow the approach to update the accessibility map and identify the impacts such as where and who would benefit more from the plan under consideration, and make the necessary adjustment.

The remainder of the paper is organized as follows. Section 2 describes the study area and some data processing issues. Section 3 explains two primary spatial analysis methods, i.e., spatial

accessibility measurement and regionalization. Other methods are considered routine and discussed briefly wherever they arise. Section 4 discusses the results and highlights the major findings of the study. Section 5 concludes the paper with policy implications and possible extensions of the work.

## 2. Study Area and Data Processing

The study area is Seoul, the capital city of South Korea, and data is mainly from the 2015 census. In 2015, approximately ten million people lived in Seoul. Insufficient public services, including childcare, have been a chronic problem for Seoul. Since launching its infrastructure plan in 2011 [36], the municipal government has been expanding PDKs, with more than 200 PDKs established annually. In 2015, PDKs accounted for about 10% of all childcare services in Seoul. The initial goal of the infrastructure plan was for PDKs to account for at least 30% of all childcare services. There is a keen interest from the municipal government and the public in gaining a better understanding of where the PDKs are located across the city and how accessible the service is for various neighborhoods.

Three datasets were used for defining spatial accessibility of PDKs: the supply and demand sides of PDK services, and the transportation networks linking them. The supply included all PDK facilities that admitted infants and children up to six years old. Nearly all kindergartens in South Korea also provide daycare, so the two are not separated in the data for PDKs. A PDK's capacity refers to the maximum number of children it can accommodate. The total number and capacity of PDKs were 1231 and 72,566 in the study area, respectively, in 2015. The locations and capacities of PDKs were based on data from the National Geographic Information Institute of South Korea (NGII) in 2015. Children between 0–6 years old defined the demand of PDKs, and the data were in 100 m × 100 m cells, also obtained from the NGII. The total number of children was 480,053 in the study area. The road network data set was obtained from the Korea Transport Institute. See Figure 1 for the spatial distributions of PDKs and children of 0–6 years old and the road networks connecting them.

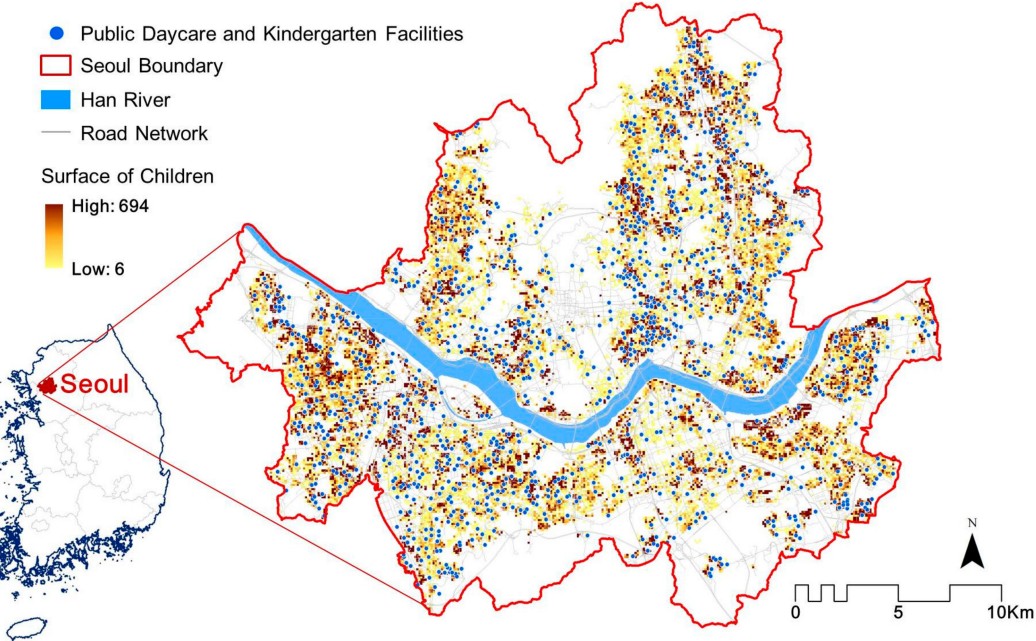

**Figure 1.** Public Daycare and Kindergarten Facilities and Children (0–6 Years) in Seoul.

While the basic population data (total count, by gender, by age group) were available in the aforementioned micro-scale cell size, other socio-economic variables such as housing, job, and education were only available in administrative area units such as Dong districts and Gu districts (hereafter referred to as Dong and Gu). Dong is the smallest administrative unit of Seoul representing various socio-economic features. Gu is a middle-sized administrative unit between Dong

and the city. Seoul has 424 Dongs and 25 Gus. Data at both Dong and Gu levels were provided by the City of Seoul. This study used data at the Dong level to define socio-economic structure, based on which homogeneous regions were constructed. The number of constructed regions was set to be 25, the same as that of Gus, to be comparable in size.

What factors capture the socio-economic structure in Seoul? Unlike the United States, there are few residents of minority ethnic groups or immigrants in South Korea. Deepening disparities in South Korea have mainly resulted from social strata over the last few decades [37], and Seoul is no exception [38]. The indicator that best describes the socio-economic level in Seoul is income. However, in South Korea, individual income data is not available, and average income data is only available in large-area units. This study considered three variables that have been widely used to study socio-economic issues in Seoul: percent of university graduates, percent of basic living security program participants, and average land value [39–41]. Educational attainment affects job outlooks and career choices, and thus, incomes. The second variable is the ratio of the population who receive subsidies for basic living security, similar to percent of people under the poverty income threshold in the US. Finally, land value is a good proxy for housing prices in highly urbanized areas such as Seoul. These three variables were extracted for the 424 Dong areas of Seoul.

## 3. Methods

### 3.1. Measuring Spatial Accessibility by the 2SFCA Method

Spatial accessibility emphasizes the relative ease by which services, PDKs in case of this analysis, can be reached by residents at a given location by overcoming the spatial separation between them. Spatial accessibility can be measured in a variety of ways, but in recent years, methods that can simultaneously account for both the demand and supply of facilities have been favored. This is to overcome the limitations of the traditional gravity model, i.e., the absence of effect by the amount of demand [33]. Among various methods, the two-step floating catchment area (2SFCA) by Luo & Wang (2003) [42] has been the most widely adopted one, as it accounts for both proximity and availability of service providers. The method is conceptually appealing because it regards capacity restrictions as well as local competitions while allowing for cross-border service-seeking behavior [16]. Due to these analytical advantages, 2SFCA is recently applied to analyze the spatial accessibility of daycare and kindergarten [15,16]. The method is convenient to implement in a GIS environment, and its result can be intuitively interpreted as the supply-demand ratio [27].

The formula of 2SFCA is written as:

$$A_i = \sum_{j \in \{d_{ij} \leq d_0\}}^{n} R_j = \sum_{j \in \{d_{ij} \leq d_0\}}^{n} \left( \frac{S_j}{\sum_{k \in \{d_{kj} \leq d_0\}}^{m} D_k} \right)$$

where $d_{ij}$ ($d_{kj}$) is the distance between demand location $i$ (or $k$) and supply location $j$, $D_k$ is the demand amount at location $k$, $S_j$ is the supply capacity at location $j$, and $n$ and $m$ are the total numbers of supply and demand locations, respectively.

In essence, the first step searches all demand locations $k$ that are within a catchment from supply location j (i.e., $d_{kj} \leq d_0$), sums up those demands and calibrates the ratio of that supply and its surrounding demands, denoted by $R_j$, as a preliminary assessment of supply availability there. The second step searches all supply locations j around each demand location $i$, and sums up all ratios $R_j$ within the catchment (i.e., $d_{ij} \leq d_0$). By doing so, the availability of any supply within the catchment from a demand location contributes to its accessibility. A larger value of $A_i$ implies better accessibility for demand location $i$. In our study, accessibility score A can be simply interpreted as available PDK seats per eligible child. To avoid small values, the scores are then inflated 1000 times and thus equivalent to several PDK seats per 1000 children.

A critical parameter for the 2SFCA method is the catchment area size $d_0$. According to Kim (2016) and Noh (2004), 700 m is suggested as an adequate catchment distance from a PDK in Seoul [21,43]. Please note that the 700 m catchment distance is measured in road network distance not Euclidean distance. In this study, based on the 2015 data, 81.5% of children (391,286 out of 480,053 children) in Seoul are in such a catchment from a PDK. One major reason for choosing a distance-based measure is that it is a short-range reachable by multiple modes (e.g., walking, biking, private vehicle or bus) while there is much uncertainty in travel time measures. Road network distances between each residential cell and each PDK were calculated by the ArcGIS Network Analyst. The automated ArcGIS toolkit provided by [44] (pp. 112–113) was used to implement the 2SFCA method.

As stated previously, the socio-economic variables such as housing price, education attainment level, and population receiving public subsidy were only available in Dong areas. It was necessary to aggregate the spatial accessibility scores at the cell level to the Dong level. There were 19,826 100 m × 100 m cells and 424 Dong areas. Accessibility in a Dong was calculated using the weighted averages of those cells within the Dong where the number of children in each cell was the weight.

*3.2. Constructing Homogenous Regions of Comparable Size by GIS-Automated Method*

One of the primary objectives of this study is to examine regional variations in access to PDKs in Seoul. Maps such as Figure 2 (in Section 4.1 below) show the variability of accessibility across administrative units such as Dong and Gu areas, and they provide some baseline for our understanding. However, our focus here is to examine the disparities in accessing PDKs across geographic areas as well as across major demographic groups. As discussed previously, we did not have access to data of individuals, and various demographic groups were usually interwoven in residential settlements. A proxy measure of disparity was used to identify various levels of subpopulation concentrations in areas and examine how accessibility varied across these areas. Here, we introduce the approach we developed to construct regions that are homogeneous in socio-economic structure and comparable in region size (i.e., number of children). While not completely free from the criticism of possible ecological fallacy [45], the variation across these regions more closely reflects actual differences in demographic composition than the administrative units do.

The study explored two GIS-automated regionalization methods: (1) Regionalization with Dynamically Constrained Agglomerative Clustering and Partitioning (REDCAP) [46,47]; and (2) Mixed-Level Regionalization (MLR) [48]. Both methods are a step forward from traditional regionalization methods with many merits, such as accounting for spatial adjacency and attribute homogeneity, attaining regions above a minimum population size, being highly automated and scale flexible, etc. [49]. As the REDCAP consistently underperformed in major indices (e.g., within-region homogeneity and compactness in region shape) in our analyses, only the results by the MLR are reported. Therefore, the remainder of the paper focuses on the MLR method.

The development of MLR is primarily for decomposing areas of large population and merging areas of the small population to derive regions that are composed of different (mixed) areal units. However, for our purpose, the MLR collapses to simply merge areas (here Dong) of the small population to generate regions of comparable population. The MLR was developed based on the modified Peano Curve algorithm (MPC) [50] and modified space-scale clustering (MSSC) [51]. The former accounts for spatial connectivity and compactness and assigns a spatial order $O_{si}$ for each Dong $i$, and the latter considers attributive homogeneity and defines its attributive order $O_{ai}$ (in our case, based on the concentrated disadvantage index or CDI value). With the spatial order $O_{si}$ and the attributive order $O_{ai}$ determined and normalized, the MLR balances the two by their corresponding weights $W_s$ and $W_a$ to define an integrated clustering order $O_i$ such as

$$O_i = W_s \cdot O_{si} + W_a \cdot O_{ai}$$

The weights are defined by the user as long as either weight is larger than 0 or their sum equals 1. In our case study, after various experiments, we chose $W_s = W_a = 0.5$ to strike a reasonable balance between compactness in shape and homogeneity in the CDI value in the derived regions.

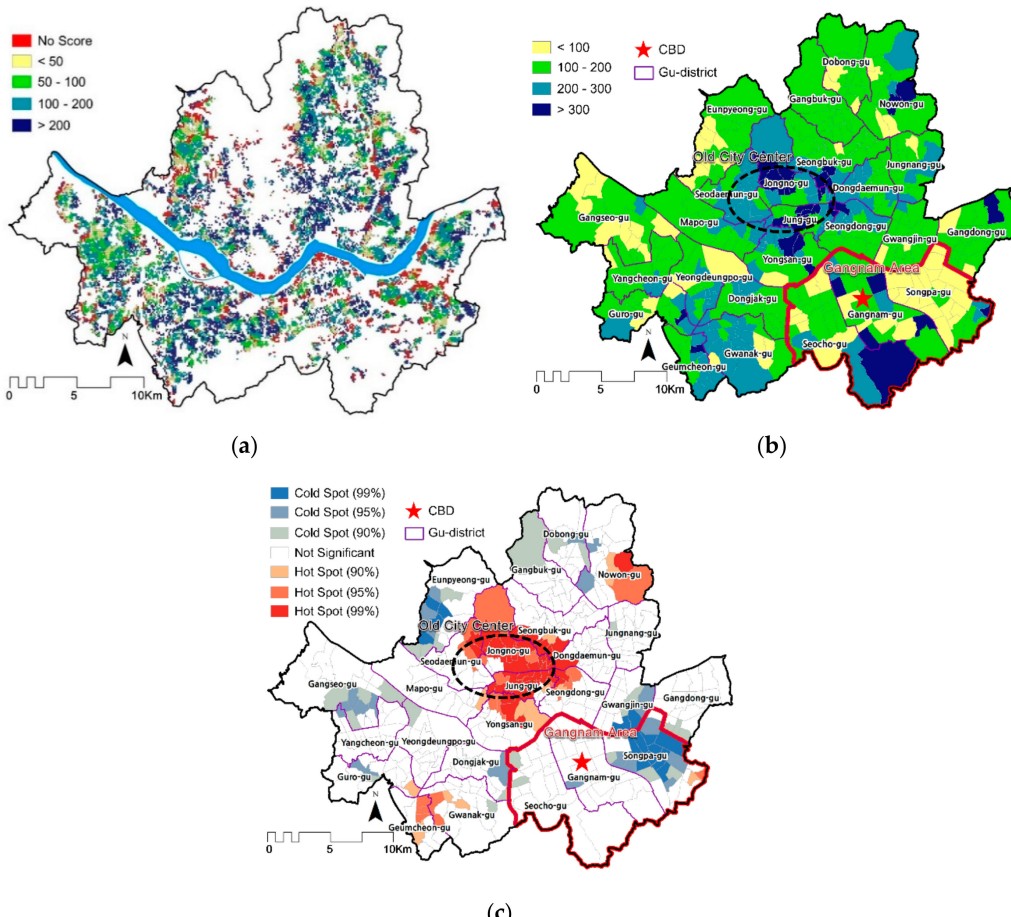

**Figure 2.** Spatial Accessibility of PDKs in Seoul: (**a**) 100 m × 100 m Cells, (**b**) Dongs, (**c**) Cold/Hot Spots.

## 4. Analysis Results

### 4.1. Spatial Accessibility across Cells and Dongs

As explained in Section 3.1, our analyses began with using the 2SFCA method to measure the spatial accessibility across 19,826 cells of size 100 m × 100 m. The following data sets were defined for implementing the 2SFCA toolkit:

(1)　A supply (point) layer of 1231 PDKs with corresponding capacities,
(2)　A demand layer of 19,826 cells with locations defined by their centroids and amounts measured by numbers of children 0–6 years old, and
(3)　A road network distance matrix between the above two within 700 m (i.e., the catchment area size).

A multiplier of 1000 was applied to the initial accessibility scores so that the values could be interpreted as the number of PDK seats per 1000 children. In the study area, a total of 480,053 children competed for 72,566 seats, yielding an average accessibility score of 151.16 seats per 1000 children. Please note that the weighted average accessibility score by the 2SFCA is about the ratio of total supply and total demand in a study area [44] pp. 110–111. This average score implied that overall, only about 15% of the children of eligible ages could be accommodated by the PDKs. The remaining demand had to be fulfilled by private providers or home care.

Figure 2a shows the spatial accessibility of PDKs across the cells. As discussed previously, daycare and kindergarten are typical examples of neighborhood facilities, and parents are sensitive to spatial impedance. It is important to use data of a fine geographic resolution in accessibility analysis of such a service with a small catchment area. This is enabled by the population data set at the cell level. The result provides an opportunity for parents to be knowledgeable of the access in their current neighborhoods or look for neighborhoods with adequate access. The information is also valuable for relevant policymakers to be aware of where the service is sufficient (or even with surplus) and where it falls short. The 100 m × 100 m cell map in Figure 2a reveals much detail and precision, but it is not very helpful for identifying major PDK accessibility patterns in Seoul. By aggregating the accessibility scores to 424 Dongs, Figure 2b smooths out some of the variability. The regional disparity is more clearly identified in the results from the 424 Dongs. The average PDK accessibility was 172, and the standard deviation was 81. The disparity was very large as the lowest accessibility was 1 while the highest was 592. The accessibility pattern at the Dong level can be further augmented by the hot-spot analysis [52], as shown in Figure 2c. Areas with excellent accessibility to PDKs are clustered in the old city center and several isolated spots on the edge of the city. The old city center used to have the most abundant PDKs before it was replaced by the so-called 'Gangnam' area (Gangnam-gu, Seocho-gu, and Songpa-gu) in 2000, where PDKs had appeared as early as the 1980s and had increased in number over time. Even though the old city center area does not currently have as many PDKs as it did in the past, we can infer that the PDK accessibility is still relatively high, due to the PDKs established and the fall in the child population.

The cold spots of PDK accessibility appear in more areas than the hot spots, but the magnitude is relatively low. The cold spots tend to cluster in the city's perimeter areas, especially in the southeast areas. In these cold spots, some are parts of Gangnam areas. Interestingly, the cold spots are clustered at a high level there since the Gangnam area is known to have the best living conditions with plentiful amenities [53] and located in a central business district (CBD). One likely explanation is that private daycare and kindergarten (PrDK) are preferred to PDK in the area, and the demand for PDK is relatively low. This speculation needs to be validated by analysis with PrDK supply data, which is not available to us. Our fieldwork reveals that about half of 250 prestigious English schools for children are clustered in Gangnam.

Based on the cell-level accessibility result, Figure 3 shows the frequency distribution of the spatial accessibility of PDKs in Seoul. It resembles a normal distribution with a relatively steeper slope on the left, as about 60% of children have below-average accessibility. The accessibility value is 0 for 1.8% of the overall children population (8637 children). In other words, no PDKs can be reached within 700 m for children in those cells. Additionally, 9.3% (44,632) of the children have accessibility below 50 (i.e., less than one-third of the average). In the meantime, 6.2% of children (29,678) enjoy the accessibility of more than 300 (i.e., twice or more than the average). The strong contrast between the haves and have nots is evident in access to PDKs in Seoul.

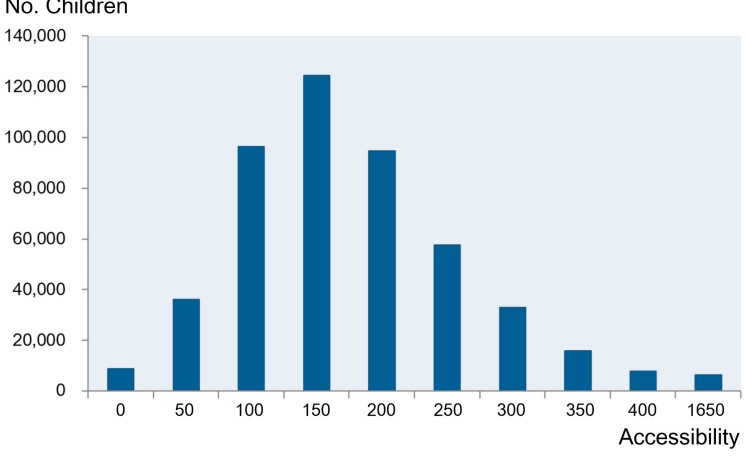

**Figure 3.** Numbers of Children across PDK Accessibility Ranges.

*4.2. Defining the Concentrated Disadvantage Index (CDI) and Constructing Regions*

Various socio-economic variables can be used as indicators of concentrated disadvantage. This study adopted three popular variables to indicate regional gaps in Seoul: (1) percent of university graduates, (2) percent of basic living security program participants, and (3) average land value. A CDI was obtained by normalizing these three variables and taking their averages. In implementing the MLR, the minimum threshold population for an aggregated region was employed to control the number of homogeneous regions. After many trials, we settled with 14,500 as the minimum number of children in a region, yielding 25 homogeneous regions that match the number of Gu regions.

The results from the constructed regions were assessed by spatial compactness and attributive homogeneity. As a spatial compactness index, the isoperimeter quotient (IQ) is defined as the ratio of the area of a region to the area of a circle with the same perimeter as the region. The higher the IQ, the more compact a region is in shape. The homogeneity is defined as the coefficient of variation (CV). The lower the CV value of a constructed region is, the higher its homogeneity is. As shown in Table 1, spatial compactness was higher in the Gu regions (mean of IQ = 0.4358) than in the constructed regions (mean of IQ = 0.2317). The CV's mean value of constructed regions was 0.3608, lower than that of Gu regions (0.4106), and thus constructed regions are more homogenous than Gu regions. The MLR method is designed to construct regions in such a way to enhance the attributive homogeneity, and thus indeed yielded a more favorable homogeneity measure in the generated regions. One likely explanation for more complex shapes in constructed regions is that the socio-economic coherence captured by the MLR is shaped more by Seoul's physical environment (e.g., topography, waterways) and road network than the administrative Gu regions. The latter might also have been districted consciously to be compact to facilitate internal connections and distribute public resources and services. Please note that one may change the values of compactness and homogeneity for the constructed regions when adjusting their corresponding weights in the MLR analysis, and the case study adopted equal weights (50–50%).

**Table 1.** Comparing MLR-constructed regions and administrative Gu regions.

|  | **Constructed Regions** | **Gu Regions** |
|---|---|---|
| Compactness (isoperimeter quotient) | | |
| Range (minimum-maximum) | 0.1041–0.5729 | 0.2576–0.6705 |
| Mean | 0.2317 | 0.4358 |
| Homogeneity (coefficient of variation) | | |
| Range (minimum-maximum) | 0.0–1.0 | 0.0–1.0 |
| Mean | 0.3608 | 0.4106 |

Note: Homogeneity is derived from standardizing CDI values from 0 to 1.

Table 2 shows basic statistics for the number of children, CDI, and accessibility of PDKs in the constructed regions and Gu regions. The number of children varied from 4812 to 37,739 across the Gu regions. By imposing a minimum threshold population of 14,500 children, the number of children ranged from 14,710 to 33,535 for the constructed regions, with much more balanced region sizes. The mean values of CDI for the two types of regions were close to 0 since CDI was based on standardized z values. The standard deviation of CDI was higher in the constructed regions. As discussed, the nature of the MLR method was to inherently enhance the homogeneity of CDI and decrease its variability within the regions while increasing the deviation of CDI across regions. The variability in the accessibility of PDKs was less in the constructed regions than the Gu regions. The patterns of CDI and accessibility were not necessarily consistent with each other, which will be discussed in more depth in Sub-Section 4.3.

**Table 2.** Basic statistics for variables in MLR-constructed regions and administrative Gu regions.

|  | Minimum | Maximum | Mean | SD |
|---|---|---|---|---|
| No. Children | | | | |
| Constructed Regions | 14,710 | 33,535 | 19,202 | 5246 |
| Gu Regions | 4812 | 37,739 | 19,202 | 7886 |
| Concentrated Disadvantage Index (CDI) | | | | |
| Constructed Regions | −1.04 | 1.22 | −0.01 | 0.60 |
| Gu Regions | −1.02 | 1.11 | 0.06 | 0.52 |
| Accessibility of PDKs | | | | |
| Constructed Regions | 102.58 | 253.09 | 160.85 | 36.53 |
| Gu Regions | 92.67 | 295.54 | 175.43 | 47.37 |

Figure 4a,b show the spatial patterns of CDI across the 25 MLR-constructed regions and the 25 Gu regions, respectively. A lower CDI value, labeled by a smaller number, corresponds to a higher socio-economic status in both maps. The two were largely consistent but differed in detail. Regions with higher CDI were found in the north, while the ones with lower CDI were in the Gangnam area and its adjacencies. Here we use region 8 in Figure 4a as an example to illustrate the value of the regionalization approach. It is known a greenbelt with a relatively low socio-economic status, where development has long been banned from controlling urban sprawls. It was detected by the MLR approach. By districting it into the three wealthiest Gu regions (labeled as 1, 2 and 3) in Figure 4b, its distinctive structure was smoothed out.

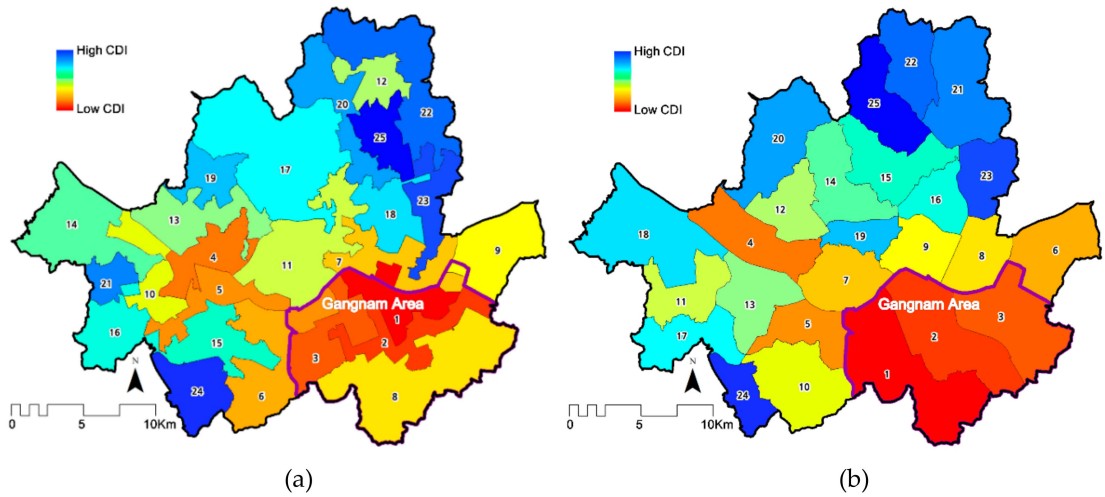

(a)　　　　　　　　　　　　　　　　　　　　　　(b)

**Figure 4.** Rankings of CDIs for (**a**) Constructed Regions, (**b**) Gu Regions. (Numbers represent CDI rankings, and a larger number means more disadvantaged socio-economic status.

### 4.3. Disparity in Accessibility across Constructed Regions

As discussed previously, the MLR-constructed regions are more coherent in socio-economic structure than the Gu regions, so this sub-section focuses the analysis of disparity in PDK accessibility across the constructed regions. Figure 5 depicts the spatial accessibility of PDKs in the constructed regions, with the numbers indicating the CDI rankings. Regions with the best accessibility of PDKs are centered in three areas: central, northeast, and southwest. On the other hand, regions with the poorest accessibility of PDKs are in the Gangnam area, the northwestern corner, and two local pockets in the north outskirts. Neither visual examination nor regression suggests any correlation between PDK accessibility and CDI.

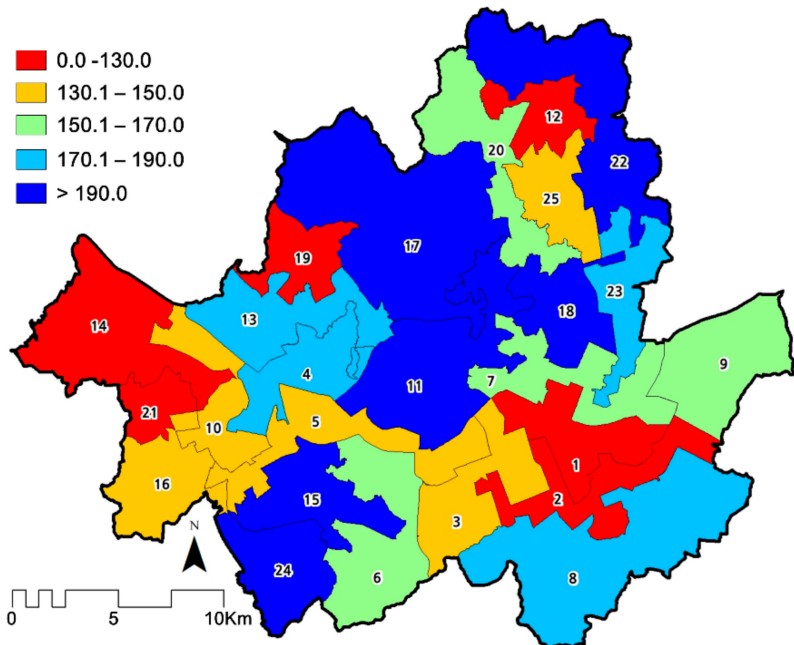

**Figure 5.** Spatial Accessibility of PDKs in Constructed Regions. (Numbers represent CDI rankings, and a larger number means more disadvantaged socio-economic status).

In the following, we classify the regions and discuss related policy implications. With reference to the average values of CDI and accessibility, the 25 regions are classified into four types:

(1)    Double-Disadvantage Regions, with above-average CDI and below-average accessibility,
(2)    Double-Advantage Regions, with below-average CDI and above-average accessibility,
(3)    Accessibility-Disadvantage Regions, with below-average CDI and below-average accessibility, and
(4)    CDI-Disadvantage Regions, with above-average CDI and above-average access.

Figure 6 shows the four types of regions. The Double-Disadvantage Regions are found in the north and around the outskirts in the west, where the development has been halted for a long time. These regions should be placed with the highest priority for the municipal government to expand the provision of PDKs there. Providing public facilities preferentially to regions with a shortage of PDK facilities and low socio-economic status is desirable for promoting social equity. On the other side, the Double-Advantage Regions are mainly located in the south of the old city center and to the west and in the south corner of the Gangnam area. These regions are relatively weathy neighborhoods and enjoy good accessibility of PDKs, and therefore any additional placement of PDKs there should not be made without well-justified causes.

The policy implications for the remaining two types of regions are less straightforward. The Accessibility-Disadvantage Regions include most of Gangnam, its northeastern neighboring area, and two local pockets in the southwest and the north outskirt. As discussed in Sub-Section 4.1, a shortage of PDKs in these areas may be offset by the availability of high-end private childcare facilities. Without a study of the accessibility of overall childcare services (private and public combined), we should be refrained from making any concrete policy suggestion. The CDI-Disadvantage Regions are mostly in the north and a small cluster in the southwest corner. The PDK accessibility of this region is above average, but its socio-economic level is not. The regions have long been plagued by poverty and other structural problems. It is remarkable that residents in these regions have become early beneficiaries from recent efforts of PDK expansions in the city. Such efforts cannot be halted as it is less feasible for local residents to seek private childcare.

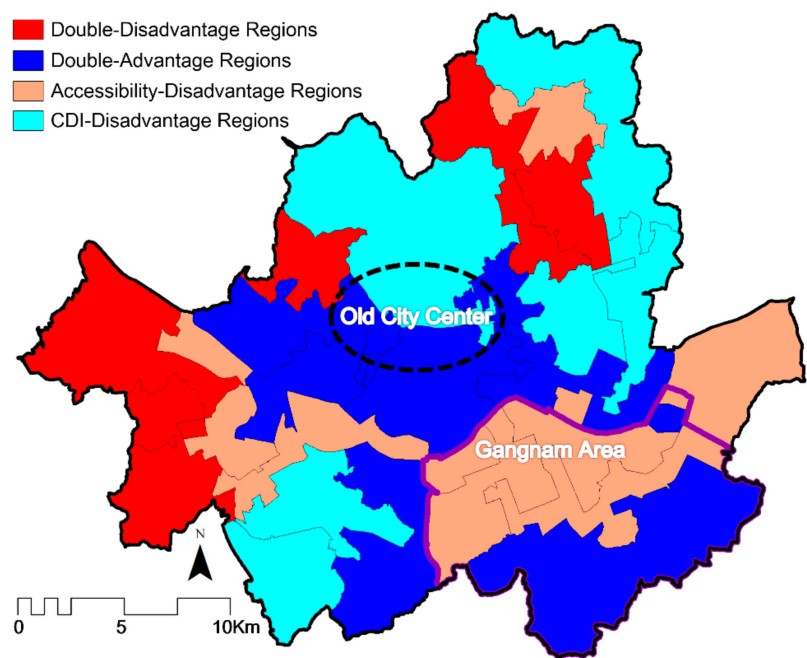

**Figure 6.** Four Regional Types derived from the Spatial Accessibility of PDKs and CDIs.

## 5. Concluding Remarks

This study analyzes the spatial accessibility of PDKs in Seoul, South Korea, and identifies regional disparity in the accessibility using GIS-constructed regions. The availability of population data at a fine geographic resolution (i.e., 100 m × 100 m cell) enables us to measure the spatial accessibility more precisely. This is important since parents, in their decision of seeking childcare, are very sensitive to the distances from these facilities. The GIS-automated regionalization method, MLR, is used to delineate homogenous regions in terms of a CDI. These constructed regions then serve as the analysis unit to assess regional disparity in PDK accessibility. Specifically, four types of regions are identified by intersecting the variables of accessibility score and CDI value. Policy implications are explored in correspondence to each type. One may order the priority of future PDK expansion in the four regions as; Double-Disadvantage Regions > CDI-Disadvantage Regions > Accessibility-Disadvantage Regions > Double-Advantage Regions.

The research can be extended in several directions. A more comprehensive measurement of CDI needs to include more variables, and other methods (in place of the current simple summation of standardized variables) may be considered to integrate those variables. Other scenarios of regionalization results, in conjunction with fieldwork, can be assessed to derive more meaningful regions for both analysis and actionable public policy implementation. As stated previously, the collection of data on private childcare facilities will complete the analysis of accessibility of overall childcare, and help us examine their distribution is complementary or in competition to the provision of PDKs. While research results change with updated data and specific methods, it is our hope that the research framework developed in this study will be adopted, amended, and improved by others for analysis of spatial disparity of accessibility in general.

**Author Contributions:** H.K. initiated the project and conducted the analysis. F.W. helped the design of research framework. H.K. and F.W. collaborated on drafting the manuscript and multiple revisions.

**Funding:** This research received no external funding.

**Acknowledgments:** Lan Mu generously provided the MLR software. We are grateful for Angela Antipova and Yujie Hu, two of the co-editors of the special issue, for inviting us to contribute and for offering guidance and comments on an earlier version. Valuable comments by two anonymous reviewers are also appreciated.

**Conflicts of Interest:** The authors declare no conflict of interest.

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
