# Peer review of "Disparity in Spatial Access to Public Daycare and Kindergarten across GIS-Constructed Regions in Seoul, South Korea"

_sustainability, doi:10.3390/su11195503_

Round 1

Reviewer 1 Report

This manuscript presents a good example of GIS application in social and urban studies. The spatial accessibility to public daycare and kindergarten (PKD) in Seoul is assessed based on a fine scale population data, and the disparity of accessibility in GIS-generated analytic regions is analyzed. I would like to suggest accept the manuscript after few minor revisions:

First, the contribution of this study and the filled gap with regards to the literature should be strengthened or more explicitly elaborated in the introduction section. Accessibility to PKD has essential meanings for public and Seoul, but other than more accurate population data and regionalization method themselves, what other contributions could be came up with, e.g. will finer scale data and spatial contiguous homogeneous lead to more unbiased policies comparing with previous studies?

Second, the rational for the utilization of 2SFCA could be improved at the beginning of section 3.1 by listing and comparing a) other accessibility assessment methods and b) empirics using 2SFCA.

Third, other trivia revision could be 1) Reference 44. Wang & Robert, 2015 is in Papers in Applied Geography not Applied Geography; 2) The abbreviation PRDK, which I think means private daycare and kindergarten, not explained at its first appearance in line 253; 3) The reason for selecting 14,500 as population threshold (line 273) should be explained; 4) Lines 336-337 are confusing. Based on the maps, regions with accessibility larger than 190 in Figure 5, like 22 and 24, have highest CDI value in Figure 4, meaning low socioeconomic status and low rank in CDI, which is contradictory to “… a high rank in CDI in general”.

Author Response

We appreciate the constructive and helpful comments raised by the reviewers. The following lists our responses. Major changes are highlighted in yellow in the manuscript.

Reviewer 1:

Highlight major contributions.

Three major contributions are outlined toward the end of section 1 (Lines 79-88 on Page 2).

Strengthen the rationale for using 2SFCA.

We elaborate more on this in section 3.1 (Lines 141-150 on Page 4)

Editorial comments.

All are excellent points and addressed accordingly.  See Lines 280-282 on Page 8 for justification for choosing 14,500 as threshold population size in MLR. We have rewritten the whole discussion related to Figures 4&5 in section 4.3

Reviewer 2 Report

The paper is interesting and relevant.

To help the reading I will put in the correct place the figures in the text.

This manuscript presents a research on the spatial accessibility to public daycare and kindergarten in Seoul.

 I suggest some minor revisions:

- in the introduction the authors have to mansion some traditional planning literature from England and United States on spatial accessibility to local public services, particularly daycare and playgrounds (5 minutes walking distance,…)

- images and table have to be in the correct part of the text to help the reader.

Author Response

We appreciate the constructive and helpful comments raised by the reviewers. The following lists our responses. Major changes are highlighted in yellow in the manuscript.

Reviewer 2:

The only major comment is on strengthening the literature review on “spatial accessibility to local public services, particularly daycare and playgrounds.” See Lines 35-46 on Pages 1-2.